# Functional and Structural Changes in Diaphragm Neuromuscular Junctions in Early Aging

**DOI:** 10.3390/ijms25168959

**Published:** 2024-08-17

**Authors:** Andrei N. Tsentsevitsky, Guzel V. Sibgatullina, Yulia G. Odoshivkina, Venera F. Khuzakhmetova, Anna R. Tokmakova, Anastasia A. Ponomareva, Vadim V. Salnikov, Guzalia F. Zakirjanova, Alexey M. Petrov, Ellya A. Bukharaeva

**Affiliations:** 1Kazan Institute of Biochemistry and Biophysics, Federal Research Center “Kazan Scientific Center of RAS”, 2/31 Lobachevsky Street, P.O. Box 30, 420111 Kazan, Russia; atsen@list.ru (A.N.T.); kam-guz@yandex.ru (G.V.S.); odnoshivkina_y@mail.ru (Y.G.O.); venerik87@mail.ru (V.F.K.); annna.tok@gmail.com (A.R.T.); na.ponomareva@mail.ru (A.A.P.); salnikov_russ@yahoo.com (V.V.S.); gffysiology@gmail.com (G.F.Z.); 2Department of Normal Physiology, Kazan State Medical University, 49 Butlerova Street, 420012 Kazan, Russia; 3Institute of Fundamental Medicine and Biology, Kazan (Volga Region) Federal University, 420008 Kazan, Russia

**Keywords:** acetylcholine, aging, contraction, diaphragm, end-plate, neurotransmitter release, neuromuscular junction, respiration, synaptic vesicle, synapsin

## Abstract

Age-related impairment of the diaphragm causes respiratory complications. Neuromuscular junction (NMJ) dysfunction can be one of the triggering events in diaphragm weaknesses in old age. Prominent structural and functional alterations in diaphragm NMJs were described in elderly rodents, but NMJ changes in middle age remain unclear. Here, we compared diaphragm muscles from young adult (3 months) and middle-aged (12 months) BALB/c mice. Microelectrode recordings, immunofluorescent staining, electron microscopy, myography, and whole-body plethysmography were used. We revealed presynaptic (i) and postsynaptic (ii) changes. The former (i) included an increase in both action potential propagation velocity and neurotransmitter release evoked by low-, moderate-, and high-frequency activity but a decrease in immunoexpression of synapsin 1 and synaptic vesicle clustering. The latter (ii) consisted of a decrease in currents via nicotinic acetylcholine receptors and the area of their distribution. These NMJ changes correlated with increased contractile responses to moderate- to high-frequency nerve activation. Additionally, we found alterations in the pattern of respiration (an increase in peak inspiratory flow and a tendency of elevation of the tidal volume), which imply increased diaphragm activity in middle-aged mice. We conclude that enhancement of neuromuscular communication (due to presynaptic mechanism) accompanied by improved contractile responses occurs in the diaphragm in early aging.

## 1. Introduction

The diaphragm muscle (DIAm) is the main respiratory muscle, which is constantly used throughout life. However, like other muscles, it is subject to age-related dysfunction and sarcopenia [1,2,3,4]. Age-related weakness and muscle fiber atrophy of DIAm may impede breathing and expulsive, non-ventilatory behaviors essential for airway clearance, thereby contributing to respiratory complications with aging [5,6,7].

Contractions of DIAm fibers are triggered by signals from phrenic motoneurons located within the ventral horn (lamina IX) of the cervical spinal cord [8]. Axonal action potentials (APs) propagate via myelinated phrenic nerves reaching presynaptic nerve terminals from which acetylcholine (ACh) is released. ACh activates muscle-type nicotinic ACh receptors (nAChRs) located at the crests of the postsynaptic folds, which form the end-plate region of the muscle fiber. Depolarizing currents via the activated nAChRs generate end-plate potentials (EPPs), whose amplitudes are usually much higher than the threshold for triggering AP on the muscle fiber. The latter finally initiates a contractile response of the muscle fiber [9]. In this way, neuromuscular junction (NMJ) translates the impulses from motoneurons to muscles fibers with a high safety factor, i.e., the amount of ACh released per nerve impulse being greater than that required to trigger an AP in the muscle fiber [10,11]. The reliability of neuromuscular communication is required to preserve motor activity under various physiological and pathological conditions [10,12,13,14].

A number of studies revealed structural and functional changes in NMJ components (end-plates, presynaptic terminals, and perisynaptic Schwann cells) in elderly rodents [15,16,17,18,19,20,21]. Age-related changes are manifested as elevations in axonal branching and end-plate fragmentation, a reduction in the neurotransmitter-filled synaptic vesicle population and nAChR function, disorganization sites of synaptic vesicle exocytosis, and partial denervation and re-innervation deficits [18,19,21,22,23,24,25,26,27,28]. All these abnormalities appeared in mice muscles after 19 months. Particularly, studies of NMJs in the DIAm of rodents have focused on alterations at moderate aging (~24 months) or older [20,24,25,29,30,31,32,33,34]. However, the impact of early aging on the DIAm NMJs is less clear. Therefore, we aimed to define structural and functional changes in DIAm NMJs of middle-aged (12 months) vs. young adult (3 months) BALB/c mice. The mean life expectancy of BALB/c mice is about 22 months [35], hence 12 months of age is slightly more than half the average lifespan of the strain. At this middle age, initial events that can further trigger NMJ dysfunction and sarcopenia can appear.

## 2. Results

### 2.1. Changes in Pre- and Postsynaptic Parameters of Neuromuscular Transmission in Early Aging

#### 2.1.1. Propagation of Nerve Impulse along the Axon

Extracellular microelectrode recording allows us to estimate the velocity of AP propagation along the motor nerve (Figure 1). Particularly, the interval between the stimulus artifact and the onset of nerve terminal AP (latency), the interval between the nerve terminal AP and postsynaptic response (synaptic delay), and the latency including synaptic delay (Lat.^S^) were calculated.

With increasing age from 3 to 12 months, latency and synaptic delay were decreased by 14 ± 20% (from 1.184 ± 0.242, n = 64 to 1.021 ± 0.238 ms, n = 33; *p* = 0.005) and 10 ± 16% (from 0.598 ± 0.114 to 0.541 ± 0.095 ms; *p* = 0.024), respectively (Figure 1A,B). An estimation of the latency in combination with the synaptic delay showed a decrease in this time by 14 ± 14% (*p* < 0.001) in 12- vs. 3-month-old mice (Figure 1B). These data mainly point to an elevation of AP propagation velocity along the motor nerve, which is myelinated almost along its entire length, excluding the distal synaptic part.

Note that in NMJs of intermediate-aged mice (9 months of age), latency and synaptic delay were not significantly different from 3-month-old mice and were 1.140 ± 0.167 ms (n = 14; *p* = 0.37) and 0.535 ± 0.107 ms (*p* = 0.14), respectively (Appendix A). Accordingly, processes triggering an increase in AP propagation velocity can occur after 9 months of age.

#### 2.1.2. Spontaneous and Evoked Neurotransmitter Release

Spontaneous ACh release causes the generation of miniature postsynaptic responses, miniature EPPs (MEPPs), whose frequency reflects the intensity of spontaneous exocytosis events (Figure 2A). MEPP frequency was similar in 3- and 12-month-old groups (0.93 ± 0.43 s^−1^, n = 67 and 1.1 ± 0.63 s^−1^, n = 37 in young and in older mice, respectively, *p* = 0.312 between groups) (Figure 2B).

Evoked exocytosis was induced by 0.5 Hz stimulation of the phrenic nerve (Figure 2A). Stimulus-induced EPPs are the result of the release of multiple quanta. The quantal content of EPPs (m) was assessed as a ratio of mean EEP and MEPP amplitudes (please see Section 4 for details). Quantal content was 21 ± 35% (*p* = 0.001) higher in 12-month vs. 3-month-old mice (21.4 ± 6.2, n = 68 vs. 25.9 ± 7.5, n = 38) (Figure 2C). Accordingly, NMJs of 12- vs. 3-month-old mice are characterized by enhanced ACh quantal release in response to nerve activation at a low frequency, whereas spontaneous exocytosis remains unaltered. Synchrony of the evoked quantal release, assessed as a ratio of EPP and MEPP rise time (RT/rt) [36], was not significantly different in 12- vs. 3-month-old groups (Figure 2D).

There were no differences in spontaneous and evoked ACh release in 9- vs. 3-month-old mice. Indeed, the MEPP frequency and quantal content of EPPs was 1.07 ± 0.47 s^−1^ (n = 14; *p* = 0.26) and 19.35 ± 5.02 (*p* = 0.25) in 9-month-old mice, respectively (Appendix A). Accordingly, the elevation of quantal content of EPPs observed in 12-month mice occurred after 9 months of age.

#### 2.1.3. Parameters of Postsynaptic Responses

The currents via nAChRs generate postsynaptic depolarization. The average amplitude of postsynaptic response to one quantum (i.e., MEPP) was 0.37 ± 0.12 mV (n = 68) and 0.28 ± 0.09 mV (n = 38) in 3- and 12-month-old mice, respectively. Accordingly, the MEPP amplitude decreased by 26 ± 23% (*p* < 0.001) in older mice (Figure 3A). Importantly, the kinetics of MEPPs, assessed as rise and decay times, were similar in 3- and 12-month-old animals (Figure 3B,C). It can be assumed that with aging, the postsynaptic membrane sensitivity to ACh decreases, leading to the reduction in MEPP amplitudes. Another possibility is that a decrease in quantal size may occur in 12-month-old mice. Note that the amplitude-temporal parameters of multi-quantal EPPs were similar in 3- and 12-month-old mice (Appendix A).

It should be noted that in 9-month-old mice, the MEPP amplitude was 0.36 ± 0.17 mV (n = 14; *p* = 0.8 as compared to 3-month-old mice) (Appendix A). This is further evidence that age-related neuromuscular changes start to appear after 9 months of age, therefore further experiments were performed on 12-month-old animals.

#### 2.1.4. Neurotransmission at Different Modes of Phrenic Nerve Activation

In physiological conditions, during quiet breathing, phrenic motoneurons discharge at 20–30 Hz, while forceful breathing requires higher frequencies (50–70 Hz) of motoneuron activity [8]. Furthermore, different frequencies of nerve activation engage different synaptic vesicle pools and recycling pathways in the NMJs [37,38,39,40,41]. Next, we tested neurotransmitter release at moderate- and high-frequency nerve stimulation in 3- and 12-month-old mice (Figure 4).

Prolonged intermittent stimulation at 20 Hz (with 0.5 s rest intervals between stimulus trains consisting of 63 stimuli each) caused a reduction in neurotransmitter release: initially fast (within the first second) and then slower (Figure 4A). Older mice were characterized by initially higher quantal content and its slow rundown during 20 Hz activity. Furthermore, the first stimulus in the new train after a 0.5 s rest interval led to the facilitation (“peak”) of neurotransmitter release as compared to the ACh release upon the last stimulus in the previous train. This seems to reflect the delivery of synaptic vesicles into the sites of exocytosis for the 0.5 s rest period, i.e., mobilization [42,43]. In older mice, these peaks of neurotransmitter release were higher than in 3-month-old mice (Figure 4A). Eventually, 45 ± 44% (*p* < 0.001) more ACh quanta were released for 3 min of 20 Hz intermittent stimulation in 12- (n = 7) vs. 3-month-old (n = 11) mice (Figure 4B).

Then, we tested neurotransmitter release in response to shot trains (consisting of 63 stimuli) at 10, 20, 50, and 70 Hz. The stimulus trains were divided by 10 s rest periods (Figure 4C). At all stimulation frequencies used, the total amount of ACh quanta released was significantly higher (by 53 ± 26%, *p* < 0.001 at 10 Hz; by 55 ± 29%, *p* < 0.001 at 20 Hz; by 61 ± 29%, *p* < 0.001 at 50 Hz; and by 62 ± 30%, *p* < 0.001 at 70 Hz) in 12- (n = 7) vs. 3-month-old (n = 11) mice (Figure 4C). Also, at all frequencies of stimulation, the quantal content decreased rapidly from an initial value to a steady-state level (plateau). The initial decrease in quantal content mainly reflects the depletion of the readily releasable pool (RRP). Assuming a “sequential model” of synaptic vesicle pool involvement in neurotransmitter release [44], RRP size can be roughly estimated by plotting quantal content (*y*-axis) against cumulative quantal content (*x*-axis) and drawing a straight line through the decay phase; its intercept with *x*-axis gives an estimate of the approximate RRP size [45,46]. This evaluation, performed at 10, 20, 50, and 70 Hz stimulus trains, showed that the size of RRP was approximately two times more (*p* < 0.001) in 12- vs. 3-month-old mice (Figure 4D). Particularly, RRP size was ~1000 and 2000 quanta in 3 and 12 months, respectively. This estimation is close to previous morphological and functional observations that predicted the size of RRP to be about 1700 vesicles in adult mouse levator auris muscle [45,47].

Thus, enhanced neurotransmitter release occurred at short moderate- to high-frequency stimulus trains and at prolonged moderate-frequency activity in older mice. This was accompanied by the increased facilitation of neurotransmitter release after short rest periods during intermittent 20 Hz stimulation. Accordingly, the enhancement of neurotransmitter quanta release can rely on the elevation of both evoked exocytosis and synaptic vesicle mobilization in the NMJs.

### 2.2. Immunofluorescent Analysis of Post- and Presynaptic Components

Specific marker of postsynaptic ACh receptor α-bungarotoxin conjugated with fluorescent dye was used to compare morphological parameters of the end-plates in NMJs of 3- and 12-month-old animals (Figure 5A). The end-plate area in 3-month-old mice was 330 ± 188 µm^2^ (143 NMJs from n = 7), while this area in 12-month-old mice was 274 ± 138 µm^2^ (102 NMJ from n = 5), which is 17% less (*p* = 0.023) than in younger animals. At the same time, the estimation of the end-plate perimeters showed an increase in the mean perimeter of 21% (from 114 ± 47 to 138 ± 61 µm; *p* < 0.001) in older mice (Figure 5B).

As presynaptic markers, antibodies against synapsin 1, a synaptic vesicle-clustering protein, and SNAP-25, an active zone protein, were applied. In all NMJs of younger and older mice, there was an exact match of presynaptic proteins to postsynaptic nAChRs (Figure 5A). This points to normal innervation in older mice. The immunoexpression of synapsin 1 was significantly reduced (by 44 ± 37%, *p* < 0.001) in 12- vs. 3-month-old mice (Figure 5C). The immunofluorescence of SNAP-25 was visibly unchanged. Hence, changes in the nAChR distribution accompanied by decreased synapsin 1 abundance can occur at the early stage of NMJ aging.

### 2.3. Electron Microscopic Evaluation of Neuromuscular Junctions and Muscle Fibers

First, cross-sections of the NMJ regions (11 regions from 3 mice in each group) were analyzed in 3- and 12-month-old mice (Figure 6). The area of presynaptic terminal cross sections (3 vs. 12 months: 10.5 ± 5.1 μm^2^ vs. 11.6 ± 5.2 μm^2^) and the number of presynaptic mitochondria (19 ± 7 vs. 18 ± 6) were similar in both groups (Appendix A), whereas synaptic vesicles were more diffusely distributed in axons of 12- vs. 3-month-old mice. The number of synaptic vesicles per nerve terminal cross-section were similar in both groups (Appendix A), but there was a tendency of the synaptic vesicle distribution to decrease in density (208 ± 54 and 180 ± 57 per μm^2^ in 3 and 12 months, respectively; *p* = 0.189) in middle-aged mice (Appendix A). The postsynaptic area looked noticeably more compact in older mice (Figure 6).

Next, muscle fibers (from three mice per group) were analyzed. The muscle fibers of the DIAm belong to the transverse striated type of musculature and have a classical structure (Figure 7a,b). In 3-month-old mice, the length of the sarcomere in the myofibrils was 2–2.2 μm (A-disc—1.5–1.7 μm; I-disc—0.4–0.5 μm). There was a distinct H-zone along the middle of the I-disc (Figure 7e). The orthodox mitochondria are round in shape and have an average matrix density and weakly defined cristae. Two populations represented this type of mitochondria: large mitochondria 1.5–2 μm in diameter located at the cell periphery (sarcoplasmic mitochondria) and smaller mitochondria 0.3–0.5 μm in diameter located chaotically between bundles of myofibrils in the I-disk region (interfibrillar mitochondria). In addition, a small number of glycogen granules are located between the fibrils. In the transverse section, myofibrils have a hexagonal arrangement and a Z-line can be distinguished (Figure 7c).

In 12-month-old mice, myofibrils consisted of sarcomeres with a length of 2.5–4 μm (A-disc 1.7–2 μm; I-disc 1.8–2 μm) (Figure 7b,f). Thus, the increase in sarcomere length was due to an increase in the size of the I-disc, so the A- and I-discs became approximately equal in length in older mice. Myofibrils were arranged in rows, tightly packed together. I- and A-discs in the Z-line were clearly expressed. Interfibrillar oval mitochondria were located in the area of the I-disk on both sides of the Z-line, parallel to each other, and had average linear dimensions of 0.2 µm along the short axis and 0.5–0.7 µm along the long axis, while cristae were well-defined, located parallel to each other (Figure 7f). In the transverse section (Figure 7d), it was clearly visible that these mitochondria have a complex spatial shape and larger dimensions. The mitochondria arranged between thin filaments formed a mitochondrial network several micrometers long. Between the fibrils, dark (osmiophilic) lipid droplets appeared with which the mitochondria were in contact. Accordingly, there was a more complex mitochondrial network of muscle fibers, the appearance of large lipid droplets, and a decrease in glycogen granules in older mice.

### 2.4. Muscle Contractions at Different Frequencies

Contractions of hemidiaphragm–phrenic nerve preparations were evoked by electrical nerve stimulation at 0.1, 10, 20, 50, and 70 Hz (Figure 8A). An increase in the frequency of stimulation led to the elevation of the maximal amplitude of contractile responses from 650 ± 213 mg at 0.1 Hz to 3364 ± 905 mg at 70 Hz (n = 11; in ~5.2 times) in 3-month-old mice and from 712 ± 225 mg at 0.1 Hz to 4482 ± 1189 mg at 70 Hz (n = 8; in ~6.3 times) in 12-month-old mice (Figure 8B).

A comparison of the DIAm contractions elicited by stimulus trains at different frequencies in 3- and 12-month-old mice revealed a significant increase in contraction amplitudes in older animals (Figure 8A,B): at 20 Hz by 29 ± 29% (*p* = 0.009) at 50 Hz by 52 ± 33% (*p* = 0.003) and at 70 Hz by 33 ± 35% (*p* = 0.056). Thus, at physiologically relevant frequencies of phrenic nerve activation, the DIAm of older mice exerted a higher contractile capacity.

### 2.5. Whole-Body Plethysmography

The DIAm is the most important muscle for breathing. The quiet breathing of 3-month-old (n = 10) and 12-month-old (n = 11) mice was evaluated in vivo using whole-body plethysmography (Figure 9). Specifically, 40–60 episodes of quiet breathing were analyzed in each mouse and averaged parameters from individual mice were used for statistical analysis.

In older mice, the breathing rate was 27% less (*p* < 0.001) (Figure 9A), whereas the tidal volume tended to increase (*p* = 0.058 after removing one highly deviant value in the 3-month-old group) (Figure 9B). As a result, minute ventilation was almost equal in both groups (12 ± 4 mL in 12 months and 13 ± 6 mL in 3 months) (Figure 9C).

The kinetics of breathing was different in older mice. Particularly, expiratory time was 58% higher (*p* < 0.001) (Figure 9D), while peak expiratory flow and peak inspiratory flow increased by 39% (*p* < 0.001) and 32% (*p* = 0.038), respectively (Figure 9E). The parameter of EF_50_ (an indicator of mid-expiratory flow rate) increased by 36% (*p* < 0.001) in 12- vs. 3-month-old mice (Figure 9F). Accordingly, the respiratory muscle activity seems to be enhanced during each cycle of breathing, which allows a decrease in the breathing rate. Such a pattern of breathing can be more effective (due to relatively more alveolar vs. dead space ventilation) and less energetically costly for mice.

## 3. Discussion

The main finding of the current study is the description of early age-related functional and structural changes in the DIAm NMJs of 12-month-old vs. 3-month-old mice. Presynaptic changes included an increase in AP propagation velocity and neurotransmitter release in response to low-, moderate-, and high-frequency activity, which was accompanied by a decrease in the immunoexpression of synapsin 1, a synaptic vesicle clustering protein. Postsynaptic alterations consisted of a decrease in currents via nAChRs and the area of nAChR distribution. These pre- and postsynaptic changes correlated with increased contractile responses to moderate- to high-frequency nerve activation. Additionally, electron microscopy revealed a more diffuse synaptic vesicle distribution in axons and a more complex mitochondrial network in muscle fibers.

Twelve months is almost the middle of a mouse’s life, and early aging signs (assessed by behavioral and locomotor tests) have been previously detected in middle-aged mice [48,49]. We hypothesize that the changes in the DIAm at this age can be important events in triggering DIAm sarcopenia, which occurs much later. Indeed, the DIAm contractile capacity, cross-sectional area of muscle fibers, and abundance of sarcoplasmic proteins (calsequestrin, sarcoplasmic reticulum Ca^2+^-ATPase) were significantly reduced in 23–24-month-old mice [5,50]. Interestingly, survival into very old age (30-month-old mice) was not associated with evidence of the progression of DIAm sarcopenia [51]. The latter suggests the existence of powerful compensatory mechanisms operating even in very aged DIAm.

NMJ studies demonstrated the fragmentation of end plates, the loss of junctional folds, a reduction in nAChR function, more complex axonal branching, the loss of synaptic vesicles, the destruction of active zones, partial denervation, and defects in re-innervation, which started after 19 months and later in rodents [18,19,21,22,23,24,25,26,27,28]. Evidently, there was no DIAm dysfunction and neuromuscular transmission failure at 12 months of age, but some adaptive and compensatory mechanisms can operate in this period to preserve and even improve the functionality of the main respiratory muscle.

Axonal-related changes in middle-aged mice. First, the decreased latency directly points to an increase in the velocity of AP propagation. Furthermore, the presynaptic AP with a shorter synaptic delay caused the postsynaptic response, suggesting a tighter coupling between membrane depolarization and exocytosis events. The evoked neurotransmitter release was enhanced in response to single stimuli and much more markedly during moderate- to high-frequency activity. The latter suggests an increase in the size of RRP and synaptic vesicle mobilization. The elevation of vesicle mobilization can also explain the increased facilitation of ACh release at the beginning of each stimulus train during prolonged intermittent stimulation at 20 Hz. The activation of β2-adrenegic receptors, which enhanced the involvement of synaptic vesicles in exocytosis, also potentiated the facilitation of neurotransmitter release during intermittent 20 Hz stimulation in diaphragm NMJs [42].

The molecular mechanism of enhanced synaptic vesicle mobilization can be related to decreased synapsin 1 abundance. This protein retains synaptic vesicles in the reluctant (non-active) pool, which is condensed in the cytoplasm of nerve terminals [52]. Electron microscopy confirmed a declusterization of synaptic vesicles in the motor axon of older mice, whereas the number of vesicles was similar in both groups. The dissociation of synapsin-form synaptic vesicles during activity facilitates vesicle mobilization [53,54,55]. In synapsin-null *Drosophila* larvae, synaptic vesicles were significantly more mobile, and the number of synaptic vesicles involved in the maintenance of neurotransmission in vivo increased by ~30% [56].

Interestingly, the synapsin 1 level also decreased in hippocampal synaptosomes of 12- vs. 3-month-old rats [57]. Synapsin 1 expression was significantly reduced in motor nerve terminals of ΔFUS(1-359) mice, a model of amyotrophic lateral sclerosis, at the onset stage of the pathology, i.e., 4–5 months of age [58]. Speculatively, deficiency in synapsin can also decrease the regenerative capacity of NMJs since synapsin can be recruited for axonal growth and the formation of new synaptic vesicle clusters [59]. In addition, the synapsin-retaining reserve pool may act as a reservoir for proteins that are essential for vesicle cycling [60].

An increase in ACh release was observed in the extensor digitorum longus and soleus muscles of very old mice (29–35 months of age) as compared to 11–13-month-old mice, whereas there were no changes in the DIAm [25,29]. The evoked ACh release increased in the DIAm of 28- vs. 10-month-old rats [31]. Potentially, enhanced ACh release can accelerate the aging of the NMJs, promoting their degeneration [61] and the disruption of synaptic vesicle morphologies and end-plate organization [62]. Moderately reducing ACh release can preserve muscle mass in old mice [63]. At the same time, the activation of postsynaptic nAChRs by quantal ACh can counteract muscle atrophy induced by denervation and enhance re-innervation [64]. Evidently, ACh release should be precisely balanced to maintain NMJ health.

Overall, the enhanced neurotransmitter release and downregulation of synapsin can represent some sort of adaptive (compensatory) response to postsynaptic alterations in older mice. Alternatively, the elevation of ACh exocytosis can trigger postsynaptic changes.

Muscle fiber-related alterations in middle-aged mice. Classical features of NMJ aging are the withdrawal of axons from some postsynaptic sites and end-plate fragmentation, which are clearly expressed after 18–20 months [28,65]. In the rat and mouse DIAm, fragmentation is accompanied by a reduction in the total end-plate area [23,24]. These changes can be reflections of denervation–regeneration/nerve terminal extension–retraction cycles [21]. At the same time, miniature postsynaptic responses to single quanta were not different in 12–14- and 26–28-month-old mice, despite the higher fragmentation in the older animals [24]. Here, we found a significant decrease in MEPP amplitudes accompanied by a reduction in the end-plate area (but an increase in its perimeter, i.e., complexity) in 12- vs. 3-month-old mice. One possibility is that a decrease in nAChR activity and the “compactization” of nAChR distribution occur in middle-age mice. Interestingly, a decrease in MEPP amplitudes was observed in the DIAm of amyotrophic lateral sclerosis model mice (SOD1-G93A and ΔFUS(1-359)) at very early stages (6–8 weeks) [58,66]. Possibly, a downregulation of nAChR activation upon spontaneously released single quanta can contribute to NMJ dysfunction in the future. Alternatively, there is an adaptation to an increase in a number of quanta released from the nerve terminals during motoneuron discharge.

It cannot be excluded that the decrease in MEPP amplitudes may be a result of decreased quantal size due to the underfilling of synaptic vesicles with ACh. Indeed, quantal size in the NMJs is subject to regulation by many signaling molecules, which can be secreted by muscle fibers, e.g., endocannabinoid 2-arachidonoylglycerol, brain-derived neurotrophic factor, and transforming growth factor beta-2 [67,68,69]. However, an increase in the MEPP amplitude was observed in the gluteus maximus muscles of 28- vs. 10-month-old mice [70]. In other studies, no differences in the amplitude of miniature postsynaptic responses were revealed in the DIAm of aged (28–30 months) mice [24,29].

Regarding muscle fiber structure, an increase in sarcomere length and the complexity of the mitochondrial network were found in older mice. Similarly, a previous electron microscopy study discovered a significant age-dependent increase in the length of sarcomeres and the number and cross-section area of mitochondria in the gastrocnemius muscle in 12- vs. 3-month-old mice [71]. The branching and elongation of mitochondria were enhanced in the gastrocnemius muscle of 22–24-month-old mice [72]. This suggests an increase in fusion and/or decrease in fission of mitochondria with age, which may be in response to stress or a compensatory mechanism to improve bioenergetics efficacy [73]. Importantly, this excessive mitochondrial fusion or fission may disrupt mitochondrial network integrity and lead to a “response” from mitochondria to the nucleus, leading to muscle atrophy and weakness [74]. Furthermore, the mitochondria network configuration influences the sarcomere and filament structure in striated muscles [75]. Speculatively, the extension of the mitochondrial network (e.g., due to increased energy needs or impaired mitochondria dynamics [76]) can contribute to an increase in sarcomere length in the DIAm of middle-aged mice. Older mice (8–15 months vs. 3–6 months) appeared to have higher mitochondrial fusion and lower autophagy in quadriceps muscles [77].

Changes in contractions and pattern of breathing. Diaphragm force contractions and transdiaphragmatic pressure, an indicator of force generated by the diaphragm in vivo, were reduced in old (23–24 months) mice [1,5,78]. Also, advanced age caused an increase in stiffness accompanied by abnormalities of metabolic enzymes and mitochondrial proteins in murine DIAm [50]. Here, we found that the contractile activity of the diaphragm was improved at frequencies corresponding to both quiet and forced breathing in 12- vs. 3-month-old mice. This is correlated with increased neurotransmitter release upon moderate- and high-frequency stimulation of the phrenic nerve. In addition, whole-body plethysmography showed an increase in peak inspiratory flow (an indicator of diaphragm function [79]) accompanied by prolongation of expiratory time and a tendency of elevation of tidal volume in 12- vs. 3-month-old mice. As a result, despite the decreased breathing rate, the minute ventilation was unchanged. These alterations in breathing can favor gas exchange in the lungs by decreasing the ratio of dead space and alveolar ventilation [80]. Given a possible increase in lung volume and compliance during the development of mice from 3 to 12 months, the observed phenotypical changes in breathing may be due to changes in lung morphology and mechanical properties. However, the lung pressure–volume relationship in mice showed a steep increase in dynamic lung compliance until 13 weeks of age, after which the pressure–volume relationship did not change markedly through 23 weeks of age [81]. Furthermore, only a small increase in inspiratory capacity per body weight and no difference in lung volume were revealed in 3- vs. 12-month-old mice [82]. Age-related structural alterations, particularly lung growth accompanied by a widening of alveolar ducts, mainly appeared between middle-aged (6–12 months) and old (18–24 months) mice [82]. At the same time, another study revealed that the greatest changes in respiratory mechanics (decrease in respiratory system resistance and increase in lung volume and dynamic compliance) occur between 2 and 6 months of age in mice [83].

The comparison of mice 3–4 months and 18–20 months of age revealed an increase in the resting breathing frequency and minute ventilation (without changes in the tidal volume) with age, and these parameters increased in young mice lacking synucleins, a model of Parkinson’s disease [84]. Possibly, the increased respiratory rate could be a compensatory mechanism to keep normal gas exchange. Note that the respiratory rate, minute ventilation, and tidal volume did not change during mouse aging from 24 to 30 months [51]. Mdx mice (6 months of age), a model of Duchenne muscular dystrophy, were characterized by decreased tidal volume, respiratory rate, minute volume, peak inspiratory, and expiratory flow as compared to wild-type mice [79].

Overall, the obtained results suggest that 12-month-old mice have enhanced neuromuscular communication (mainly due to presynaptic mechanism), which correlates with improved DIAm contractions and, possibly, changes in a pattern of breathing. Hypothetically, the changes in DIAm activity can trigger mitochondrial and sarcomere remodeling in the muscle fibers. According to this scenario, presynaptic alterations in synaptic vesicle clustering and synapsin 1 expression can contribute to age-related phenomena in the main respiratory muscle.

## 4. Materials and Methods

**Animals and preparations.** Experiments were performed on isolated mouse phrenic nerve–hemidiaphragm preparations. BALB/c mice were maintained at a 12 h light/12 h dark cycle; water and food were provided ad libitum. Young, sexually mature, 3-month-old mice (33 ± 4 g), adult 9-month-old mice (29 ± 7 g), and middle-aged 12-month-old mice (37 ± 8 g) were used in the study.

The DIAm with phrenic nerve stubs was dissected into two nerve-muscle (hemidiaphragm) preparations. The preparation was placed in an experimental chamber (total volume of 5 mL) with a Sylgard-coated transparent base, and the motor nerve was loosely drawn into a suction electrode connected to an isolated stimulator. The chamber was continuously perfused at ~3 mL/min and the solution level in the chamber was kept constant. The recording solution was a modified Krebs–Ringer Solution (HEPES-buffered): 150.0 mM NaCl, 5.0 mM KCl, 2.0 mM CaCl_2_, 1.0 mM MgCl_2_, 11.0 mM glucose, and 5.0 mM HEPES, pH 7.4 with NaOH. The temperature was maintained at 22.0 ± 0.3 °C. The muscle contractions were blocked by incubating the preparations with muscle-type Nav1.4 channel blocker GIIIB μ-conotoxin (Peptide Institute Inc., Osaka, Japan) at 0.5 μM for 15–20 min followed by thorough rinsing with toxin-free buffer solution [43].

**Electrophysiological Recordings.** The methods of extracellular recordings, data acquisition, and processing have been previously described in detail [85]. Briefly, MEPPs, presynaptic APs, and EPPs were acquired in the junctional region. Extracellular saline-filled micropipettes with a tip diameter of 2–3 μm and input resistance of 2–3 MΩ were placed under the microscope (Olympus BX51WI) control in the junctional zone, where three-phase presynaptic APs were observed. The precise position of micropipettes and the stability of presynaptic AP amplitude and duration were monitored throughout the experiment. The acquired signals were filtered between 0.03 Hz and 10 kHz, digitized at 3 μs intervals by an analog-to-digital converter, fed into the computer, and analyzed. The MEPP and EPP amplitude, rise time (between 20 and 80% of maximum amplitude), and time constant of the exponential decay (tau), as well as a ratio of EPP-to-MEPP rise time (Rt/rt), were estimated. For MEPPs, the signal-to-noise ratio was >4.5:1 and the threshold for spontaneous signal detection was set at the level of 0.2 mV. The frequency of MEPPs was evaluated in each experiment after recording 150–200 spontaneous signals. As an indicator of evoked neurotransmitter release (m), a ratio of the mean EPP and MEPP amplitudes was used. For this analysis, 150–200 MEPPs and EPPs were recorded. The motor nerve was stimulated with supra-threshold stimuli with a 0.1 ms duration at a frequency of 0.5, 10, 20, 50, or 70 Hz.

**Electron microscopy of the diaphragm muscle.** The samples of the DIAm from 3- and 12-month-old mice (3 per group) were fixed with a solution of 2.5% glutaraldehyde in 0.1 M phosphate buffer (pH 7.4) at room temperature for 3 h, and then fixed with 1.0% osmium tetroxide for 1 h at room temperature supplemented with sucrose (25 mg/mL). Samples were dehydrated in a graduated series of alcohols, acetone, and propylene oxide. The samples were impregnated with epoxy resin for three days with a gradual increase in the resin concentration. The polymerization of samples in resin was carried out for three days at 37.0, 45.0, and 60.0 °C for 24 h, respectively. Ultrathin sections (thickness of 50–100 nm) of the samples were prepared on an ultramicrotome LKBIII (Stockholm, Sweden) using a diamond knife. Sections were mounted on copper grids supported by formvar and stained with saturated aqueous uranyl acetate and lead citrate. The specimens were examined using transmission electron microscopy HT7800 (Hitachi, Tokyo, Japan) at an operating voltage of 80 kV. The quantitative analysis of neuromuscular junctions (NMJs) was performed as described by Wokke et al. [86].

**Immunohistochemical analysis.** Tissue samples were fixed in a 4% paraformaldehyde solution for 1 h at room temperature. After the fixation procedure, the samples were washed three times in phosphate buffer saline for 10 min each and incubated in 0.5% Triton X-100 solution for 30 min. Then, the samples were kept for 20 min in a blocking solution containing 1% bovine serum albumin, 5% normal goat serum, and 0.5% Triton X-100 in phosphate buffer saline. In the next step, the preparations were incubated with primary antibodies against synapsin 1 (1:200; Abcam, Cambridge, UK, ab64581) and SNAP 25 (1:200; Abcam, ab31281) for 16 h at 4 °C. Then, they were rinsed three times in phosphate buffer saline for 30 min. The next step was incubation for 1 h with secondary antibodies conjugated to Alexa-647 (1:250, Invitrogen, Carlsbad, CA, USA) and Alexa-488 (1:250, Invitrogen, CA, USA) and then for 30 min with tetramethylrhodamine isothiocyanate–α-bungarotoxin (20 µg/mL; Sigma, St Louis, MO, USA) to enable the visualization of postsynaptic nAChRs. Then, the samples were washed in phosphate buffer saline three times and mounted using Surgipath SubX media (Leica Microsystems, Wetzlar, Germany). Observations were performed using the laser confocal scanning microscope Leica SP5 TCS (Leica Microsystems, Wetzlar, Germany).

The determination of relative fluorescence was performed as described by Blottner et al. [87]. Immunofluorescence images were scanned with a confocal laser scanning microscope Leica SP5 TCS at the standardized image settings, and all digitalized images were analyzed using the Leica confocal software (LasX v. 3.3.0.16799). The area pixel intensity of the defined regions of interest (ROIs) was measured in digital confocal image scans and expressed as arbitrary units (a.u.s). Corrections to the background fluorescence were made for each image.

**Analysis of contractions.** The isometric contractions of neuromuscular preparations of the diaphragm were recorded using a 4-channel Tissue Bath System (BIOPAC Systems, Inc., Goleta, CA, USA). One end of the isolated preparation was tied to a fixed hook and the other end was connected to a strain gauge with a sensitivity of 0–25 g [88]. The preparation was placed in a perfused bath (volume of 20 mL), and the nerve placed between the platinum electrodes was stimulated by supramaximal electrical stimuli of 0.2 ms duration (indirect stimulation). Resting tension of the hemidiaphragm was initially adjusted to obtain maximal contractile responses and then equilibrated for 20 min before the onset of recording. During this 20 min period, the hemidiaphragms were stimulated (one impulse per minute) to reach a steady-state level of single contractions. In the experiments, contractile responses were elicited by low-frequency stimulation (0.1 Hz) for 10 min or stimulus trains (consisting of 40 stimuli) at 10, 20, 50, and 70 Hz. The force of contractions was determined in mg. Signals were recorded and processed using AcqKnowledge software (v.4.1).

**Whole-body plethysmography.** A whole-body plethysmograph (Shanghai Tow Intelligent Technology Co., Ltd. Shanghai China) was used according to the manufacturer’s instruction and as recommended in [89,90]. Mice were placed in a closed main chamber and differences in pressure between the main and reference chambers were measured to obtain the specific parameters of breathing. The respiratory activity of mice in a conscious state and under unrestrained conditions was recorded after their adaptation for 40–60 min and only when the mice were in a calm state exhibiting quiet breathing. The monitoring of quiet breathing occurred for one hour, and 3 s segments (at least 40) without sniffing and grooming were analyzed. All chambers were cleaned thoroughly before and after each mouse. The parameters selected for the assessment of possible breathing differences (breathing rate, tidal volume, minute ventilation, expiratory time, peak expiratory flow, and peak inspiratory flow) were calculated using the WBP software(ResMass, v.1.4.2.8).

**Statistics.** Origin Pro software (v. 7.5) was used for statistical analyses. Data are presented as mean ± standard deviation (SD). The sample size (n) is the number of independent experiments on separate nerve–muscle preparations (muscles) from individual mice; n is indicated in each figure legend. There were no exclusions of outliers; the sample size was determined based on a reasonable value of SD. Normality (the Shapiro–Wilk test) and variance homogeneity (the two-sample F-test for variance) were tested. Significance was assessed by the Mann–Whitney U-test. * *p* < 0.05, ** *p* < 0.01 and *** *p* < 0.001 were considered statistically significant.

## Figures and Tables

**Figure 1 ijms-25-08959-f001:**
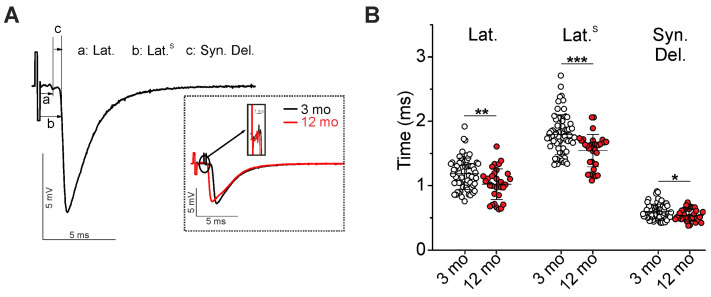
Measurements of latency and synaptic delay. (**A**) Typical extracellular recording, including artifact, AP, and EPP, is shown. Intervals corresponding to latencies (Lat. and Lat.^S^) and synaptic delay (Syn. Del.) are denoted as a, b, and c, respectively. Insert, the representative recordings from NMJs of 3- and 12-month-old mice (3 mo and 12 mo). (**B**) Quantification of latencies and synaptic delay. Data are represented as mean ± standard deviation (SD). n = 64 and 33 for 3 and 12 months. * *p* < 0.05, ** *p* < 0.01 *** *p* < 0.001 by Mann–Whitney U-test between groups.

**Figure 2 ijms-25-08959-f002:**
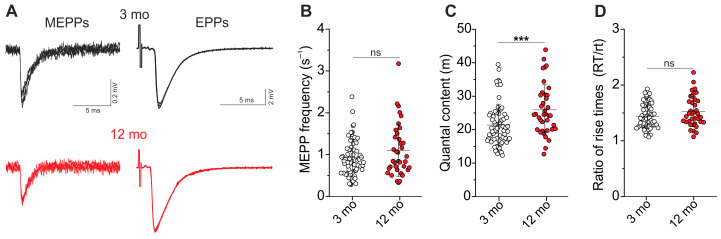
Spontaneous and evoked ACh release. (**A**) Typical traces of MEPPs and EPPs recorded extracellularly in 3- and 12-month-old mice (3 mo and 12 mo). Three traces are overlaid in each example. Quantification of MEPP frequency (**B**), quantal content (m) of EPPs (**C**), and ratio (Rt/rt) of MEPP and EPP rise times (**D**). Data are represented as mean ± SD. n = 67–68 and 37–38 for 3 and 12 months. *** *p* < 0.001 by Mann–Whitney U-test between groups; ns—non-significant.

**Figure 3 ijms-25-08959-f003:**
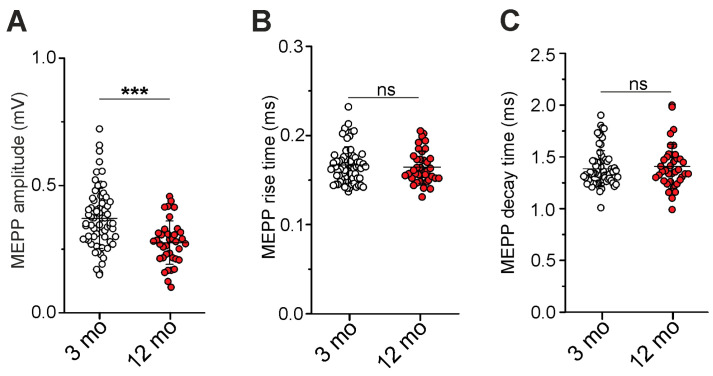
Parameters of miniature postsynaptic responses. (**A**) Amplitude, (**B**) rise time, and (**C**) decay time of MEPPs. n = 68 and 38 for 3 and 12 months. Data are represented as mean ± SD. *** *p* < 0.001 by Mann–Whitney U-test between groups; ns—non-significant.

**Figure 4 ijms-25-08959-f004:**
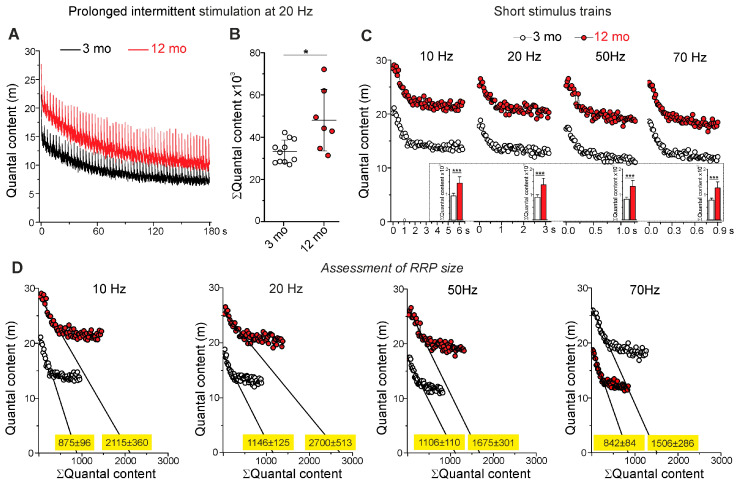
Evoked ACh release at different modes of phrenic nerve activation. (**A**) Changes in quantal content during 3 min intermittent stimulation at 20 Hz. Averaged curves are shown. (**B**) Cumulative quantal content indicating a number of quanta released for 3 min of intermittent stimulation at 20 Hz. (**C**) Dynamic neurotransmitter release upon shot stimulus trains at 10, 20, 50, and 70 Hz. Insert, graphs show the number of quanta released during these stimulus trains. (**D**) Estimation of readily releasable pool (RRP) size by plotting quantal content (m; the *y*-axis) over the cumulative quantal content (*x*-axis) and extrapolating the initial linear decay phase on *x*-axis. Data are represented as mean ± SD. n = 11 and 7 for 3 and 12 months. * *p* < 0.05 and *** *p* < 0.001 by Mann–Whitney U-test between groups.

**Figure 5 ijms-25-08959-f005:**
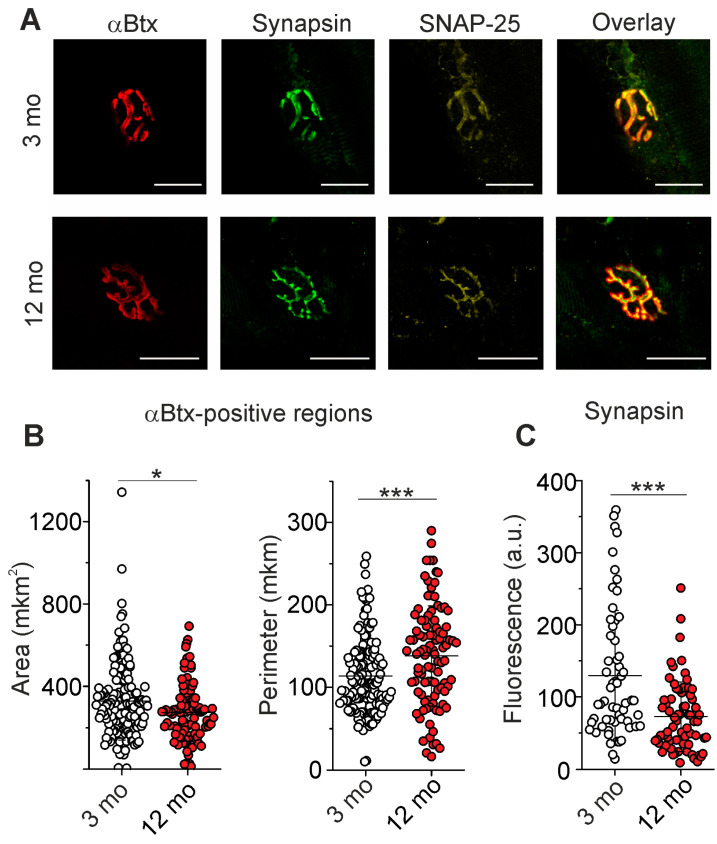
Labeling of postsynaptic and presynaptic components of NMJs. (**A**) The representative fluorescence images of NMJs, where postsynaptic nAChRs and presynaptic proteins (synapsin 1 and anti-SNAP-25) were labeled with αBtx and specific antibodies, respectively. Scale bars—25 µm. (**B**) Estimation of αBtx-positive area and perimeter. (**C**) Quantification of synapsin 1 immunofluorescence. Data are represented as mean ± SD; 143 NMJs from n = 7 (3 months) and 102 NMJ from n = 5 (12 months) were analyzed. * *p* < 0.05 and *** *p* < 0.001 by Mann–Whitney U-test between groups.

**Figure 6 ijms-25-08959-f006:**
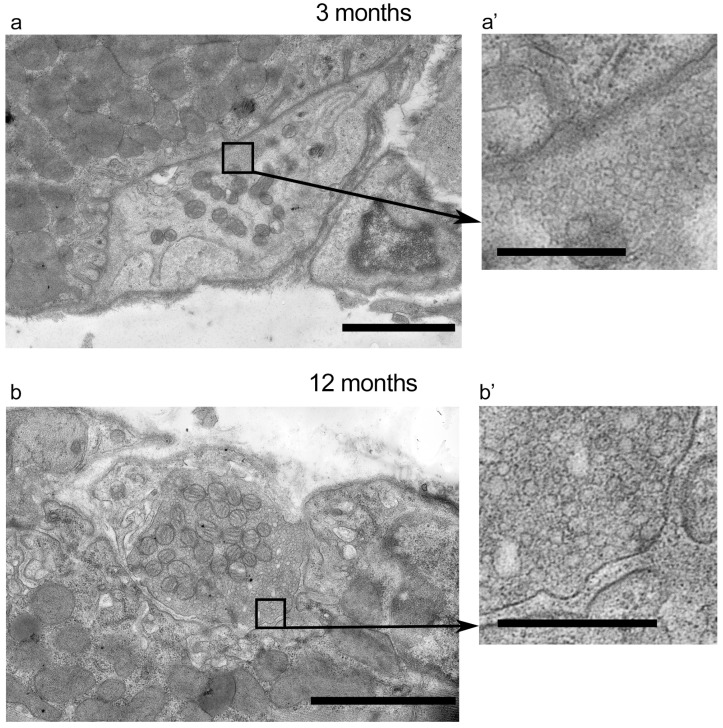
Electron microscopy of NMJs. (**a**,**b**) Representative TEM images of NMJs from 3-month-old and 12-month-old mice. (**a’**,**b’**) Regions containing active zone, synaptic vesicle cluster, postsynaptic infoldings, and synaptic cleft at a higher magnification. Scale bars—0.5 µm (**a’**,**b’**) and 2 µm (**a**,**b**).

**Figure 7 ijms-25-08959-f007:**
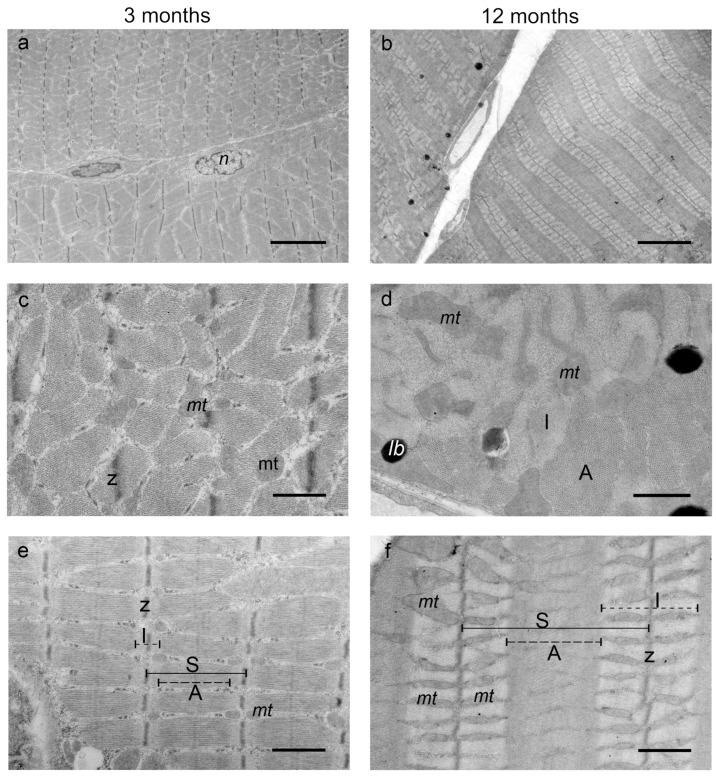
Ultrastructure of DIAm myofibrils of 3-month-old and 12-month-old mice. TEM images: (**a**,**b**) general view of muscle fiber bundles; (**c**,**d**) cross-section of myofibrils; (**e**,**f**) longitudinal section of myofibrils, sarcomere structure. Scale bars—5 µm (**a**,**b**) and 1 µm other microphotographs. Abbreviations: A—A-band, I—I-band, lb—lipid body, mt—mitochondria, n—nucleus, S—sarcomere, Z—Z-line.

**Figure 8 ijms-25-08959-f008:**
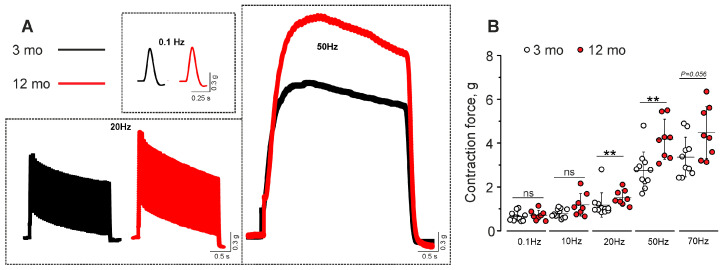
Muscle contractions evoked by nerve activation. (**A**) Representative traces of contractile responses to low- (0.1 Hz), moderate- (20 Hz), and high- (70 Hz) frequency stimulation in 3- and 12-month-old mice. (**B**) Quantification of maximal contraction force elicited by phrenic nerve stimulation at 0.1, 10, 20, 50, and 70 Hz. Data are represented as mean ± SD. n = 11 and 8 for 3 and 12 months. ns—non-significant and ** *p* < 0.01 by Mann–Whitney U-test between groups.

**Figure 9 ijms-25-08959-f009:**
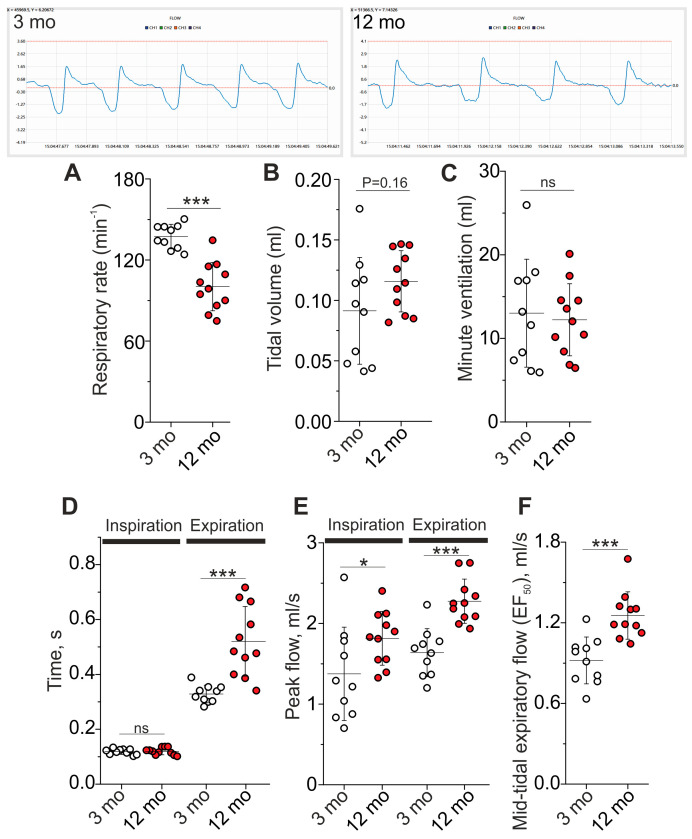
Whole-body plethysmography of 3- and 12-month-old mice. Top, the representative recordings. Quantification of respiratory rate (**A**), tidal volume (**B**), and minute ventilation (**C**) are shown. (**D**–**F**) Graphs show the kinetic parameters of breathing: inspiratory and expiratory time (**D**), peak inspiratory and expiratory flows (**E**), and mid-tidal expiratory flow or EF_50_ (**F**). Data are represented as mean ± SD. n = 10 and 11 for 3 and 12 months. * *p* < 0.05 and *** *p* < 0.001 by Mann–Whitney U-test between groups; ns, non-significant.

## Data Availability

Data used in this study are available from the corresponding author E.A.B. upon reasonable request.

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
