# Peer review of "Functional and Structural Changes in Diaphragm Neuromuscular Junctions in Early Aging"

_ijms, 2024, doi:10.3390/ijms25168959_

Round 1

Reviewer 1 Report

Comments and Suggestions for Authors

In this work, Tsentsevitsky et al. assayed the NMJ function and breath behavior of 3 month and 12 month aged mice and found increased quantal content, DIAm contractile force, and TV. Rather than focusing on aged animals, the authors studied these changes in “middle” age and fulfilled an important time point during NMJ aging. The data presented in this paper is carefully analyzed and well explained. I would suggest a few changes before considering a publication.

1.       The authors found a decrease of mEPP at 12 month old and claimed that this is due to a decrease of postsynaptic sensitivity. However, another possible cause would be the change of presynaptic vesicle size or the amount of ACh of each vesicle. The authors did not distinguish these two possibilities, though they briefly discussed in the discussion section. I suggest using FM dyes to measure the synaptic vesicle size.

2.       Line 157-158, the authors showed a higher synaptic vesicle mobilization in 12 month old animals. Another important measure of presynaptic function, termed “readily releasable pool”, can be calculated based on the data presented. Therefore, rather than simply showing a sum quantal content, the authors should estimate the RRP size to better illustrate the change of presynaptic properties. I appreciate the discussion section where they suggest this change of RRP could be due to the decrease of Synapsin.

3.       The authors claimed that the presynaptic changes are keys to DIAm contraction in line 408-413. However, their evidence is not enough to support this claim. The authors assayed these NMJ parameters and muscle/breath function. This can only be considered as “correlation” but not “causal”. To prove their claim, for example, they should mutate synapsin in 3 month old animals and see if mutant animals recapitulate the changes of 12 month animals.

4.       It is very interesting to think about the tighter NMJ transmission, lower respiratory rate but higher TV, and how this compensatory change happens. The authors argue that the increased NMJ transmission leads to a change of DIAm contractile force and breathing pattern. However, considering the increase of lung volume during 3m to 12m development, TV should increase and thus the RR may decrease. In this case, the phenotypical change of breathing is not due to the neurological change. I would suggest the authors either design experiments to distinguish these two possibilities (as mentioned above), or clarify in the discussion section.

Minor:

1.       Figure 1, the authors should index the figures by letter A,B,C… rather than “left” and “right”.

2.       The authors mentioned data of 9 month old animals but never showed the data. Since the authors already did this set of experiment on 9 month old mice, they should include the data along with 3 and 12 month old data.

3.       Given that the authors already acquired the EM images, a quantification of synaptic vesicle number and distance to presynaptic membrane should be included in Figure 6.

Reviewer 2 Report

Comments and Suggestions for Authors

This study is interesting and explains in a clear and concise manner the early age-related functional and structural changes in the DIAm NMJs of 12-month-old vs 3-month-old mice and have important contribution for the area. The study is easy to read, and objectives were correct described. The authors used different types of techniques to demonstrate them, and the figures are well represented. However, some improvements could help make the work even better:

1-     Abstract section could be more structured in accordance with the journal´s guidelines, information about animals, methods, etc.

2-     Results section: 2.1. Changes in pre- and postsynaptic parameters of neuromuscular transmission in early aging 67 2.1.1. Propagation of nerve impulse along the axon 68 Extracellular microelectrode recording allows to estimate the velocity of AP propa- 69 gation along the motor nerve (Figure 1) [36,37]. References can be eliminated in this section.
